# Domain Adaptation via Distribution and Representation Matching: A Case Study on Training Data Selection via Reinforcement Learning

## Abstract

Supervised models suffer from domain shifting where distribution mismatch across domains greatly affect model performance. Particularly, noise scattered in each domain has played a crucial role in representing such distribution, especially in various natural language processing (NLP) tasks. In addressing this issue, training data selection (TDS) has been proven to be a prospective way to train supervised models with higher performance and efficiency. Following the TDS methodology, in this paper, we propose a general data selection framework with representation learning and distribution matching simultaneously for domain adaptation on neural models. In doing so, we formulate TDS as a novel selection process based on a learned selection distribution from the input data, which is produced by a trainable selection distribution generator (SDG) that is optimized by reinforcement learning (RL). Then, the model trained by the selected data not only predicts the target domain data in a specific task, but also provides input for the value function of the RL, which exactly helps update the parameters of SDG in a policy gradient way. Experiments are conducted on three typical NLP tasks, namely, part-of-speech tagging, dependency parsing, and sentiment analysis. Results demonstrate the validity and effectiveness of our approach.

## 1 Introduction

Learning with massive data suffers from "Pyrrhic victory" where huge amounts of resource, e.g., computation, annotation, storage, etc., are consumed with many issues. One well-known issue for data-driven approaches is that data quality considerably affects the performance of learned models. Especially in NLP, such phenomenon is incredibly significant where noise and inaccurate annotations are demolishing models' robustness when applying them to various scenarios (Akopyan & Khashba, 2017; Zhang et al., 2016), one of which is cross-domain NLP tasks. In this scenario, many data and their labels are considered noisy and inaccurate when they are used across domains, resulting in the mismatch of distribution between training and test data. To reduce the impact of the noise in data and annotations, reasonable selection of training samples is proved to be an useful strategy that can help supervised models achieving equivalent performance with less computational efforts (Fan et al., 2017; Feng et al., 2018). Particularly, it is also confirmed that appropriate data selection strategy can benefit domain adaptation (Ruder & Plank, 2017) by preventing negative transfer (Rosenstein et al., 2005) from irrelevant data and noisy labels, meanwhile providing a more efficient learning process compared with other domain adaptation methods such as sample reweighing (Borgwardt et al., 2006), feature distribution matching (Tzeng et al., 2014) and representation learning (Csurka, 2017).

Although various data selection-based domain adaptation approaches were proposed for NLP tasks (III, 2007; Blitzer et al., 2007; Søgaard, 2011), most of them only consider scoring or ranking learning samples under a certain metric over the entire dataset, then select the top $n$ (or a proportion, which is usually a predefined hyper-parameter) items to learn. However, such pre-designed metrics is not always able to cover effective characteristics for transferring domain knowledge and meanwhile suffers from their inflexibility. Even though some methods learn measures across domains and models (Ruder & Plank, 2017), their hyper-parameter setting, such as the $n$, still demands further explorations. Fortunately, recent studies confirmed that RL is an effective solution for data selection in training neural networks with improving their efficiency (Fan et al., 2017) and accuracy (Feng et al., 2018). A natural extension of using RL for data selection can be considered for the cross-domain scenario.

Yet, to apply RL to data selection for domain adaptation is not a straightforward mission as those RL application for in-domain tasks. To tackle this mission, in this paper, we propose a general data selection approach for domain adaptation via RL that jointly learns data representations and matches them from the source and target domain. There are two major components in our approach: a selection distribution generator (SDG) and a task-specific predictor with a feature extractor and a classifier[1]. The SDG and the predictor are pipelined by taking each others' output as input and optimized accordingly via RL. As a result, in our approach, two major problems in domain adaptation are addressed: some redundant or noisy data across domains gives little or negative contribution to adaptation; useful information from the source domain is usually not properly organized and exploited for the target domain. Experimental results from three NLP tasks, namely, part-of-speech (POS) tagging, dependency parsing and sentiment analysis, illustrate that our approach achieve state-of-the-art performance, which confirm the validity and effectiveness of our approach.

## 2 THE APPROACH

Our approach is in the line of the unsupervised domain adaptation (Roark & Bacchiani, 2003), where one has no access to any labels of target domain data. Particularly, for a task $\mathcal{T}$, we take labeled data instances for the task from a source domain $\mathcal{D}_S$ as the input to our approach, and some unlabeled data from a target domain $\mathcal{D}_T$ as the guidance. The approach is then expected to generate an optimal selected data set and a model trained on the set for $\mathcal{T}$. An overview illustration of the entire architecture for our approach is demonstrated in Figure 1. In detail, for each step, the key component for selection, SDG, produces a distribution vector based on the representation of the selected source data from last step, then data instances are selected according to the vector and new reward is generated for next round data selection. To update the SDG, different measurements can be used to assess the discrepancy between the representations of the selected source data and the guidance set and then approximates the value function for updating. For the predictor, the loss from the classifier on the domain-specific task is used to guide the parameter updating for the feature extractor and the classifier. The details is unfolded in the following subsections, in which we firstly give the details of the two major components and then the joint training algorithm.

### 2.1 THE PREDICTOR

The predictor is the main component to perform a particular task $\mathcal{T}$. In our approach we decompose the predictor into two parts, the feature extractor and a classifier, and use them separately. The feature extractor serves as the representation learning module that transform input data to dense representations, while the classifier takes the representations for $\mathcal{T}$ in the target domain. In doing so, the predictor is a neural model in this study so that the aforementioned separation can be done. Therefore, the feature extractor is normally the first n-1 layers of the predictor given that it has n layers in total. As a result, the classifier is the last layer dedicated for model output.

**The Feature Extractor** Data in its original form, especially as natural language, is usually difficult to be exploited in neural computation. The feature extractor is thus serving as a critical component in our approach to transform the data into distributed representations for their efficient use and computation. There are two-way inputs for the feature extractor. One is the guidance set $X_g^T = \{x_1^T, x_2^T, ..., x_m^T\}$, a collection of unlabeled data drawn from the target domain, serving as the reference for the selection process. The other input is the selected data from the source domain in a "data bag", which is a batch of a certain amount of samples to facilitate the selection in this study. Let $X_S = \{x_1^S, x_2^S, ..., x_n^S\}$, $\forall x_i^S \in \mathcal{X}_S$ denote the data from the source domain, we uniformly partitions the entire data set into $N$ disjoint data bags marked as $\{B_1, B_2, ..., B_N\}$, with $B_j = \{x_{(j-1)n/N+1}^S, x_{(j-1)n/N+2}^S, ..., x_{jn/N}^S\}, j \in \{1, 2, ..., N\}$. Through the feature extractor, the outputs of the guidance set and the selected data are two collections of distribution vectors.[2]

**The Classifier** Once data selection is done, the classifier is combined with the feature extractor and trained on the selected data for a particular task. Note that the training is conducted in the source domain and the output of the classifier is used to generate gradients for back-propagation. Then the parameters of the classifier and the feature extractor are updated accordingly (with a learning rate $\beta$).

### 2.2 THE SELECTION DISTRIBUTION GENERATOR

A multi-layer perceptron (MLP) model is used as the SDG, which learns the selection policy optimized by the reward from the representations of the guidance set and the selected data by RL. In doing so, at each time step, the SDG is fed by a collection of data representations for a data bag from the feature extractor. We denote the collection as $\mathbf{\Phi}_{B_j} = \{r_1^j, r_2^j, ..., r_{|B_j|}^j\}$, where $r_l^j$

---

[1]It is not necessarily a classifier, e.g., such as a tagger. However we use the term classifier for simplicity.
[2]The formal description of such representations are given in the next section.

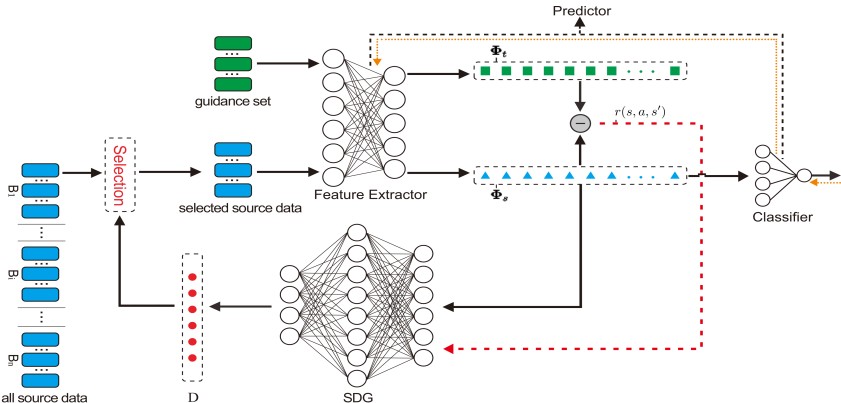

Figure 1: The architecture of a predictor with a selection distribution generator. All black solid arrows refer to data flow, while the red dashed arrow denotes reward and the orange dotted arrows indicating back-propagation of gradients from training the predictor.

$(l = 1, 2, ..., |B_j|)$ is the vector of the $l$-th sample[3] in $B_j$.[4] Then $\mathbf{\Phi}_{B_j}$ is mapped by the SDG to a vector $\mathbf{D}_{B_j} = (p_1^j, p_2^j, ..., p_{|B_j|}^j)$, $p_l^j(l = 1, 2, ..., |B_j|)$, which represents the probability for each sample that whether it should be selected individually. To learn the SDG, each $\mathbf{\Phi}_{B_j}$ is measured with $\mathbf{\Phi_t}$ to give the reward in an RL framework, which is described as follows.

**RL Components in Learning the SDG**

- **State** $(s_1, s_2, ...s_j, ...s_N)$ includes a collection of states for all $j$ with respect to $N$ data bags, where each $s_j$ indicates a status including data bag $\hat{B}_j$ and parameters of the feature extractor for the selected instances $\hat{B}_j$ from $B_j$. For simplicity we use $\mathbf{\Phi}_{\hat{B}_j}$ and $\mathbf{\Phi_t}$ to represent state $s_j$.
- **Action** For each state, the action space $A$ is a 0-1 judgment to decide if selecting an instance (1) or not (0). An action $a = \{a_k\}_{k=1}^{|B_j|} \in \{0, 1\}^{|B_j|}$, which is obtained from $\mathbf{D}_{\hat{B}_j}$. After each action, data selection gives new $\mathbf{\Phi}_{\hat{B}_j}$, then transforms state $s$ into $s'$. Thus we define the policy as $P_{\mathbf{W}}(a|s)$.
- **Reward** Since the data selection process is to ensure that the selected data fitting the distribution of the target domain, we set a reward $r(s, a, s')$ to assess the distance between $\mathbf{\Phi}_{\hat{B}_j}$ and $\mathbf{\Phi}_t$ in the current state $(s')$ and its previous state $(s)$:

$$r(s, a, s') = d(\mathbf{\Phi}_{\hat{B}_{j-1}}^s, \mathbf{\Phi}_t^s) - \gamma d(\mathbf{\Phi}_{\hat{B}_j}^{s'}, \mathbf{\Phi}_t^{s'}) \tag{1}$$

where $d(\cdot, \cdot)$ is a distribution discrepancy measurement, which can be implemented by different information-bearing functions. $\gamma \in (0, 1)$ is a discounting constant that decreases the impact from future distribution difference. Note that Eq. (1) is conducted in a sequence based on two adjacent data bags $B_{j-1}$ and $B_j$, of which $\mathbf{\Phi}_{\hat{B}_j}^{s'}$ are impacted by $\mathbf{\Phi}_{\hat{B}_{j-1}}^s$ via parameters $\Psi$ of the feature extractor updated by $\hat{B}_{j-1}$. Consequently, the state transition probability $\mathcal{P}(s'|s, a)$ is determined by stochastic optimization and other randomness in training, e.g., dropout (Srivastava et al., 2014). When better instances are selected, the reward is then expected to produce a higher value because the measurement for the previous state $d(\mathbf{\Phi}_{\hat{B}_{j-1}}^s, \mathbf{\Phi}_t^s)$ supposes to give a larger distance between $\mathbf{\Phi}_{\hat{B}_{j-1}}$ and $\mathbf{\Phi}_t$ than that for the current state.

**Distribution Discrepancy Measurement** To measure each $\hat{B}_j$ and the $X_g^T$, let $P$ be the element-wise average of $\Phi_{\hat{B}_j}$ and $Q$ the average of $\mathbf{\Phi}_t$, we use the following measurements for $d(\cdot, \cdot)$ in this paper:

- **JS**: The Jensen-Shannon divergence (Lin, 1991). $d(P, Q) = \frac{1}{2}[D_{KL}(P||M) + D_{KL}(Q||M)]$ where $D_{KL}(P||Q) = \sum_{i=1}^n p_i \log \frac{p_i}{q_i}$, with $M = \frac{1}{2}(P + Q)$.
- **MMD**: The maximum mean discrepancy (Borgwardt et al., 2006). $d(P, Q) = \|P - Q\|$.
- **RÉNYI**: The symmetric Rényi divergence (Rényi, 1961). $d(P, Q) = \frac{1}{2}[Ry(P, M) + Ry(Q, M)]$, $Ry(P, Q) = \frac{1}{\alpha-1} \log(\sum_{i=1}^n \frac{p_i^\alpha}{q_i^{\alpha-1}})$. We set $\alpha$=0.99 following Van Asch & Daelemans (2010).

---

[3]Representations in the collection follow the same order of their corresponding data instances in the bag.
[4]Similarly, the representation collection of the guidance set is denoted as $\mathbf{\Phi_t}$.

---

**Algorithm 1:** Joint training algorithm of our approach

---

**Input:** Epochs $L$; Training data $\mathbf{B} = \{B_1, B_2, ..., B_N\}$. $\mathbf{W}$ (SDG), $\mathbf{\Psi}$ (Predictor), $\mathbf{\Theta}$ (feature extractor); Loss function of the predictor $F(\mathbf{\Psi}, \hat{B}_j)$; $n_J$; $d(\cdot, \cdot)$; $\gamma$.

**Output:** Updated $\mathbf{W}$ and $\mathbf{\Psi}$ ($\mathbf{\Theta}$).

Initialize $\mathbf{W}$ and $\mathbf{\Psi}(\mathbf{\Theta})$ with standard Gaussian distribution;

**for** *epoch l = 1 to L* **do**
 $\Sigma = 0$;
 **for** *k = 1 to $n_J$* **do**
  $\Sigma_r = 0$;
  Shuffle $\{B_1, B_2, ..., B_N\}$;
  **for** *each $B_j \in \mathbf{B}$* **do**
   $\mathbf{\Phi}_{B_j}^{s_j} \leftarrow \mathbf{\Theta}_{j-1}(B_j)$; $\mathbf{\Phi}_t^{s_j} \leftarrow \mathbf{\Theta}_{j-1}(X_g^T)$;
   On current bag state $s_j$, $\mathbf{D}_j \leftarrow \mathbf{W}(\mathbf{\Phi}_t^{s_j})$; select $\mathbf{\Phi}_{\hat{B}_j}^{s_j}$ from $\mathbf{\Phi}_{B_j}^{s_j}$ via $\mathbf{D}_j$ (take action $a_j$);
   $r(s_{j-1}, a_j, s_j) \leftarrow d(\mathbf{\Phi}_{\hat{B}_{j-1}}^{s_{j-1}}, \mathbf{\Phi}_t^{s_{j-1}}) - \gamma d(\mathbf{\Phi}_{B_j}^{s_j}, \mathbf{\Phi}_t^{s_j})$
   $\Sigma_r \leftarrow \Sigma_r + \gamma^{j-1} r(s_{j-1}, a_j, s_j)$
   $\mathbf{\Psi} \leftarrow \mathbf{\Psi} - \beta \nabla_{\mathbf{\Psi}} F(\mathbf{\Psi}, \hat{B}_j)$; ( $\mathbf{\Theta}_j \leftarrow \mathbf{\Theta}_{j-1} - \beta \nabla_{\mathbf{\Theta}} F(\mathbf{\Psi}, \hat{B}_j)$ );
  **end**
  $\Sigma \leftarrow \Sigma + \sum_{j=1}^{N} \nabla_{\mathbf{W}} \log \pi_{\mathbf{W}}(a_j^k | s_j^k) \Sigma_r$;
 **end**
 $\nabla_{\mathbf{W}} \widetilde{J}(\mathbf{W}) \leftarrow \frac{1}{n_J} \Sigma$;
 $\mathbf{W} \leftarrow \mathbf{W} + \tau \nabla_{\mathbf{W}} \widetilde{J}(\mathbf{W})$;
**end**

---

- **LOSS**: The guidance loss. $d = -\frac{1}{m} \sum_{i=1}^{m} \sum_{y_t \in \mathcal{Y}_t} y_t \log p_\Phi(y_t | x_i^T)$ where $y_t$ is the label of instance $t$ from the guidance set, and $p_\phi$ the learned conditional probability of the predictor. Note that, different from aforementioned measurements, LOSS requires labels from the target domain, thus is only set as a comparison to our approach with using annotated data through the predictor.

**Optimization** The following object is optimized to obtain the optimal distribution generation policy.

$$J(\mathbf{W}) = \mathbb{E}_{P_{\mathbf{W}}(a|s)} [\sum_{j=1}^{N} \gamma^{j-1} r(s_j, a_j)] \tag{2}$$

Then the parameters of the SDG, i.e., $\mathbf{W}$, is updated via policy gradient (Sutton et al., 1999) by

$$\mathbf{W} \leftarrow \mathbf{W} + \tau \nabla_{\mathbf{W}} \widetilde{J}(\mathbf{W}) \tag{3}$$

where $\tau$ is the discounting learning rate[5], the gradient $\nabla_{\mathbf{W}} J(\mathbf{W})$ is approximated by $\nabla_{\mathbf{W}} \widetilde{J}(\mathbf{W}) = \frac{1}{n_J} \sum_{k=1}^{n_J} \sum_{j=1}^{N} \nabla_{\mathbf{W}} \log \pi_{\mathbf{W}}(a_j^k | s_j^k) \sum_{j=1}^{N} \gamma^{j-1} r(s_j^k, a_j^k)$, where $j$ refers to the $j$-th step (corresponding to the $j$-th data bag) in RL; $i$ denotes the $i$-th selection process to estimate $\nabla_{\mathbf{W}} J(W)$. Note that $\nabla_{\mathbf{W}} J(W)$ is estimated after $n_J$ times of selection over all $N$ data bags, where $n_J$ is a predefined hyper-parameter.

## 2.3 JOINT TRAINING

We jointly train the SDG and the predictor with policy gradient method (Sutton et al., 1999), which favors actions with high rewards from better selected samples. The entire joint training process is described in Algorithm 1. Following Feng et al. (2018), we choose the whole $B_j$ for the predictor to compute $r(s_{j-1}, a_j, s_j)$ when $\hat{B}_j = \varnothing$ at time step $j$ to help exclude noisy bags.

## 3 EXPERIMENT

To evaluate our approach, we conduct experiments on three typical NLP tasks: POS tagging, dependency parsing, and sentiment analysis. Details about our experiments are described as follows.

## 3.1 DATASETS

Two popular datasets are used in our experiments. For POS tagging and dependency parsing, we use the dataset from the SANCL 2012 shared task (Petrov & McDonald, 2012), with six different domains. For sentiment analysis, we use the product review dataset from Blitzer et al. (2006), with

---

[5]$\tau$ and the aforementioned $\beta$ can be self-adapted by the optimizer, such as Adam (Kingma & Ba, 2014).

| TASK | POS TAGGING/DEPENDENCY PARSING | | | | | | SENTIMENT ANALYSIS | | | |
|------|------|------|------|------|------|------|------|------|------|------|
| DOMAIN | A | EM | N | R | WB | WSJ | B | D | K | E |
| LBL. | 3.5K | 4.9K | 2.4K | 3.8K | 2.0K | 3.0K | 2K | 2K | 2K | 2K |
| UNL. | 27K | 1,194K | 1,000K | 1,965K | 525K | 30K | 4.5K | 3.6K | 5.7K | 5.9K |

Table 1: The statistics of all datasets used in our experiments, with the number presenting labeled (LBL.) or unlabeled (UNL.) samples in each domain. The domain abbreviations in different tasks are explained as follows. A:Answer, EM:Email, N:News, R:Reviews, WB:Weblogs, WSJ:Wall Street Journal, and B:Book, D:DVD, K:Kitchen, E:Electronics.

four domains. Note that for all datasets, there exists both labeled and unlabeled samples. The statistics and the domains for the aforementioned datasets are reported in Table 1.

## 3.2 COMMON SETTINGS

A major difference between our approach and other data selection methods is that the number of instances to be selected, $n$, is not fixed in our approach. Instead, it chooses the most effective one automatically. For fair comparison, we record the resulted $n$ from our approach in different tasks and use it in other methods to guide their selection. In all experiments, the source domain includes all labeled data from the dataset except that for the target domain, i.e., we take turns selecting a domain as the target domain, and use the union of the rest as the source domain. $N$ is set according to a uniform bag size of around 1K. For the guidance set, we follow Ruder & Plank (2017) and randomly select half of the instances from all the test data in the target domain discarding their labels.

Considering that the starting reward needs to be calculated from a reliable feature extractor, before the regular training, we pre-train the predictor on all source data for 2 epochs, then initialize parameters of SDG with Gaussian variables. Later the predictor and SDG follows ordinary updating paradigm in each training epoch. In all experiments, we use Adam (Kingma & Ba, 2014) as the optimizer, and set $\gamma$ to 0.99 following Fan et al. (2017) and $n_J$ to 3. All the deep learning based methods are configured with the same neural networks as our corresponding experiments.

|  | A | EM | N | R | WB | WSJ |
|--|------|------|------|------|------|------|
| JS-E | 93.16 | 93.77 | 94.29 | 93.32 | 94.92 | 94.08 |
| JS-D | 92.25 | 93.43 | 93.54 | 92.84 | 94.45 | 93.32 |
| T-S | 93.59 | 94.65 | 94.76 | 93.92 | 95.32 | 94.44 |
| TO-S | 93.36 | 94.65 | 94.43 | 94.65 | 94.03 | 94.22 |
| T+TO-S | 94.33 | 92.55 | 93.96 | 93.94 | 94.51 | 94.98 |
| T-S+D | 93.64 | 94.21 | 93.57 | 93.86 | 95.33 | 93.84 |
| TO-S+D | 94.02 | 94.33 | 94.62 | 94.19 | 94.93 | 94.67 |
| MDAN | 94.83 | 95.69 | 95.97 | 95.14 | 95.88 | 95.23 |
| RANDOM | 92.76 | 93.43 | 93.75 | 92.62 | 93.53 | 92.68 |
| ALL | 95.16 | 95.90 | 95.90 | 95.03 | 95.79 | 95.64 |
| SDG (JS) | 95.37 | 95.45 | 96.23 | 95.64 | 96.19 | 95.74 |
| SDG (MMD) | **95.75** | 96.23 | 96.40 | 95.51 | **96.95** | 96.12 |
| SDG (RÉNYI) | 95.52 | **96.31** | **96.62** | **95.97** | 96.75 | **96.35** |
| SDG (LOSS) | 95.46 | 95.77 | 95.92 | 95.50 | 96.03 | 95.82 |

## 3.3 POS TAGGING

Table 2: POS tagging results (accuracy %) for each domain.

**The Predictor** We use the Bi-LSTM tagger proposed in Plank et al. (2016) as the predictor.

**Baselines** Following Ruder & Plank (2017), we compare our approach to five baselines: 1) **JS-E**: top instances selected using Jensen-Shannon divergence. 2) **JS-D**: top instances selected from the most similar source domain, where the similarity between domains are determined by Jensen-Shannon divergence. 3) Bayesian optimization (Brochu et al., 2010) with the following settings: **T-S**, term distribution similarity; **TO-S**, topic distribution similarity; **T+TO-S**, joint term and topic distribution similarity; **T-S+D**, term distribution similarity and diversity; **TO-S+D**, topic distribution similarity and diversity. 4) **MDAN**: a recent multiple source domain adaptation with adversarial learning (Zhao et al., 2018). 5) **RANDOM**: a random selection model that selects the same number of instances according to our proposed approach. 6) **ALL**: The predictor is trained on all available source data.

**Results** POS tagging results are reported in Table 2. Overall, our approach with different distribution discrepancy metrics outperforms all baselines based on the same feature extractor. This observation demonstrates the excellent adaptability of our approach on this tasks with complicated structural variance in sentences. Among the four metrics, Rényi divergence achieve the best overall performance, which is slightly surpassed by MMD in the ANSWER and WEBLOGS domain. It is observed that the convergence of our approach results in around 50% of the source data selected.

### 3.4 DEPENDENCY PARSING

**The Predictor** The Bi-LSTM parser proposed by Kiperwasser & Goldberg (2016) is the predictor.

**Baselines** For dependency parsing, we use the same baselines introduced in the POS tagging task.

**Results** The performance (labeled attachment scores, LAS) of dependency parsing is reported in Table 3. Similar to POS tagging, MDAN stand out among the baselines and the term distribution-based method (T-S) as well as its combination with diversity features (T-S+D) surpass other Bayesian optimization baselines, while our models outperform the measurement-based as well as neural network based ones significantly in most domains. However, different from POS tagging, in this task,

|  | **A** | **EM** | **N** | **R** | **WB** | **WSJ** |
|---|---|---|---|---|---|---|
| JS-E | 81.02 | 80.53 | 83.25 | 84.66 | 85.36 | 82.43 |
| JS-D | 82.80 | 79.93 | 81.77 | 83.98 | 83.44 | 80.61 |
| T-S | 83.79 | 81.09 | 82.68 | 84.66 | 84.85 | 82.57 |
| TO-S | 82.87 | 81.43 | 82.07 | 83.98 | 84.98 | 82.90 |
| T+TO-S | 82.87 | 81.13 | 82.97 | 84.65 | 84.43 | 82.43 |
| T-S+D | 83.72 | 81.60 | 82.80 | 84.62 | 85.44 | 82.87 |
| TO-S+D | 82.60 | 80.83 | 84.04 | 84.45 | 85.89 | 82.33 |
| MDAN | 83.56 | 84.45 | 84.12 | 84.96 | 85.38 | 84.91 |
| RANDOM | 81.28 | 83.41 | 81.03 | 82.67 | 82.46 | 80.74 |
| ALL | **85.65** | **87.78** | **86.07** | **87.27** | 85.51 | 85.56 |
| SDG (JS) | 84.03 | 85.98 | 84.17 | 86.25 | **86.22** | 85.24 |
| SDG (MMD) | 84.19 | 86.25 | 84.87 | 86.80 | 85.57 | 84.37 |
| SDG (RÉNYI) | 84.55 | 85.11 | 85.27 | 86.93 | 85.65 | **85.79** |
| SDG (LOSS) | 83.97 | 85.86 | 84.05 | 86.21 | 86.03 | 84.98 |

Table 3: Dependency parsing results (LAS) for each domain.

the predictor trained on the entire source data still performs the best on some domains, which can be explained by the complexity of the task. Yet, our models, e.g., the SDG (JS) and SDG (RÉNYI), outperform the ALL model in the last two domains, with only half of the source data used.

### 3.5 SENTIMENT ANALYSIS

**The Predictor** We use the CNN classifier proposed by Kim (2014) as the predictor in this task.

**Baselines** In addition to the baselines used in POS tagging and dependency parsing, we use a series of extra baselines from previous studies: 1) **SCL**, the structural correspondence learning proposed by Blitzer et al. (2006); 2) **SST**, the sentiment sensitive thesaurus method (Bollegala et al., 2011); 3) **DAM**, a general-purpose multiple source domain adaptation method proposed by Mansour et al. (2008); 4) **SDAMS-LS** and **SDAMS-SVM**, the specially designed sentiment domain adaptation approach (Wu & Huang, 2016) for multiple sources with square loss and hinge loss, respectively.

**Results** Table 4 presents the results of sentiment analysis. Similar to previous tasks, it is observed that our approach still performs competitively in this task, even though compared with the algorithms dedicatedly designed for sentiment analysis (e.g., SDAMS) and a general deep learning based multiple source domain adaptation method (MDAN). A potential reason for our weaker re-

|  | **B** | **D** | **E** | **K** |
|---|---|---|---|---|
| JS-E | 72.49 | 68.21 | 76.78 | 77.54 |
| JS-D | 75.28 | 73.75 | 72.53 | 80.05 |
| T-S | 75.39 | 76.27 | 81.91 | 83.41 |
| TO-S | 76.07 | 75.92 | 81.69 | 83.06 |
| T+TO-S | 75.75 | 76.62 | 81.74 | 83.39 |
| T-S+D | 76.20 | 77.60 | 82.66 | 84.98 |
| TO-S+D | 77.16 | 79.00 | 81.92 | 84.29 |
| MDAN | 78.65 | 80.72 | **84.46** | 86.04 |
| RANDOM | 76.78 | 75.28 | 78.25 | 82.27 |
| ALL | 78.48 | 79.68 | 80.58 | 84.50 |
| SCL | 74.57 | 76.30 | 78.93 | 82.07 |
| SST | 76.32 | 78.77 | 83.57 | 85.19 |
| DAM | 75.61 | 77.57 | 82.79 | 84.23 |
| SDAMS-LS | 77.95 | 78.80 | 83.98 | 85.96 |
| SDAMS-SVM | 77.86 | 79.02 | 84.18 | 85.78 |
| SDG (JS) | 79.37 | 81.06 | 82.38 | 85.78 |
| SDG (MMD) | 79.57 | 81.08 | 82.68 | 85.69 |
| SDG (RÉNYI) | **80.07** | **82.07** | 82.28 | **86.18** |
| SDG (LOSS) | 79.57 | 80.58 | 81.88 | 85.08 |

Table 4: Sentiment analysis results (accuracy %).

sults on Electronics target domain is that dedicatedly designed methods include relation graphs among key words as prior knowledge, while our model aims for a wider application without task-specific consideration. Slightly different from previous tasks, in this task, around 40% source data are selected upon the convergence of our models.

### 3.6 DISCUSSION

In all three tasks, our approach achieve the best overall performance when there are half or less than half source data selected to train the predictor. The comparisons between our approach and the basic distribution measure-based methods, the general-purpose multiple source approach as well as the previous models (in sentiment analysis) across all tasks illustrate the effectiveness of our approach in selecting the most useful instances for the target domain while eliminating the noise. However,

domain variance still plays an important role affecting model performance, where it is also task-specific. Compared to POS tagging and dependency parsing, in sentiment analysis there exists more significant bias across domains, e.g., "small" could be positive in one domain but negative in another.

Consequently, topic related domains express similar sentiment expressions. The investigation of the selected data indicates that our approach chooses more instances from the similar domains in sentiment analysis (e.g., BOOK ⇒ DVD), while the selected instances in POS tagging and dependency parsing are more balanced across domains. This observation suggests the effectiveness of our approach in adapting different tasks with the most appropriate strategy.

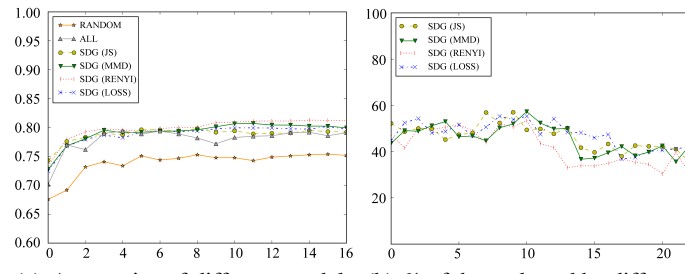

(a) Accuracies of different models against training epochs.

(b) % of data selected by different models against training epochs.

Figure 2: Investigation of using different models on the DVD domain for sentiment classification.

In addition, there still exist side effects while leveraging noise filtering and relevant instance selection, which can be observed from the slightly weaker results on Electronics domain in sentiment analysis and the fact that our approach is outperformed by training on all of the sources in parsing task. Such phenomenon implies that filtering some irrelevant instances may lose intrinsic beneficial information for target domain classification, and policy gradient method with partial data sometimes converges to a local optima given structured data with many indirect relations.

## 4 ABLATION STUDIES

### 4.1 PERFORMANCE AND EFFICIENCY OF DIFFERENT MODELS

To better understand the behavior of our model with different measurements, we investigate their performance using a case study on the DVD domain in sentiment analysis. We record the accuracy curves of different models with respect to their training epochs, as shown in Figure 2(a). In general, our models show similar performance and are significantly better than the RANDOM one. Interestingly, their curves are also similar to the ALL model but shows a much stable performance with the epoch increasing. This observation demonstrates that there exist severe noise when using all source data directly, while our models are able to overcome such limitation.

Another investigation is to study how much data are selected by different measurements in our approach. We display the number of instances selected by the four measurements in Figure 2(b), using the same domain and task setting as that in Figure 2(a). Overall our models with different measurements share similar behavior in selecting source data in terms of selection numbers. They tend to select more data at the beginning stage, i.e., before 10 epochs, then reduces selected instances to a smaller set and keeps the performance of the predictor (comparing with the curve in Figure 2(a)). Among all measurements, Rényi divergence tend to select less data instances while achieving a better performance when we match the numbers with the results reported in Table 4.

### 4.2 DISTRIBUTION VISUALIZATION

To better demonstrate the effect of our approach, we use the sentiment analysis for the DVD domain with SDG (RÉNYI) for visualized comparison among the distributions of instances in different scenarios. Following Tzeng et al. (2014), we plot in Figure 3 the t-SNE visualizations of the feature vectors learned from the feature extractor in four settings: vectors before training (learning with initialized weights) (Figure 3(a)), trained with SDG (Figure 3(b)), trained with JS-E (Figure 3(c)), and directly trained on all source data (Figure 3(d)). It is observed that, in original vectors, DVD and BOOKS are similar, while ELECTRONICS and KITCHEN are different from them as well as to each other. When trained with all source data, feature vectors could show some change in their distributions where instances from different domains are mixed and closer to the target domain. In the case where JS divergence is minimized for each domain, we can see a further mixture with closer representation matching. However, their domain adaptation capability is limited since the vectors are not optimized for the target domain. On the contrary, when trained with our approach, the selected instances result in a highly similar distribution as that in the target domain, as shown in Figure 3(b) the matched shape between the points in red and other colors, which is more beneficial than the structure in non-RL based embeddings (3(c)). Consider that it is a dynamic selection process in our

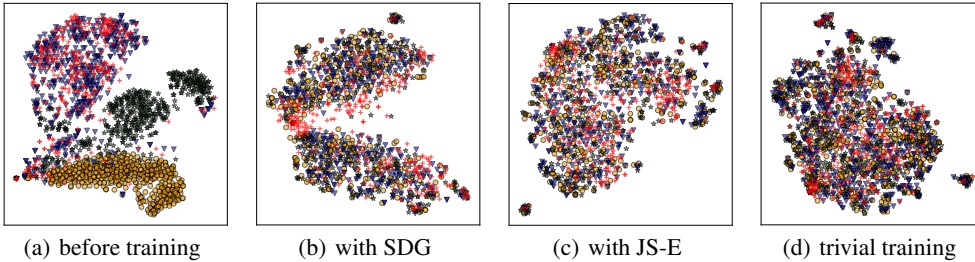

| (a) before training | (b) with SDG | (c) with JS-E | (d) trivial training |

Figure 3: t-SNE visualization of learned features for sentiment analysis on the DVD domain in different settings. Red cross, blue triangle, green star, and orange circle represent DVD, BOOKS, ELECTRONICS, and KITCHEN (best viewed in color with enlarged figures).

approach, the observation from Figure 3 confirms that such process ensures that the selected data could shape a target domain alike distribution without redundant instances.

## 5 RELATED WORK

Many studies have been conducted recently for domain adaptation with neural networks (Long et al., 2015; 2016; Shu et al., 2018; Shankar et al., 2018). Their methodologies include representation learning (Glorot et al., 2011; Chen et al., 2012; Baktashmotlagh et al., 2013; Zhao et al., 2018), reweighing samples from the source domain (Borgwardt et al., 2006; III, 2007; Gong et al., 2013), and seeking a way for explicit feature space transformation (Gopalan et al., 2011; Pan et al., 2011; Long et al., 2013), etc.

Normally, the transferable knowledge across domains are derived from some particular data, while others contribute less and are costly to be learned from (Axelrod et al., 2011; Ruder & Plank, 2017). Thus, previous studies conduct domain adaptation through selecting relative and informative source data according to the nature of the target domain, such as entropy-based source data selection (Song et al., 2012), Bayesian optimization based selection (Ruder & Plank, 2017), etc. Especially in various NLP tasks, Training data selection has been widely used, such as multilingual NER (Murthy et al., 2018), machine translation (Chen et al., 2016; van der Wees et al., 2017), and language model (Moore & Lewis, 2010). Consider that data selection in general is a combinatorics' optimization problem with exponential complexity, it thus needs to be reduced to a series of sequential decisions, while RL provides a good solution in this case. For example, in Feng et al. (2018), they used sequential one-step actions for each single instance where every action is decided based on the previous one. In doing so, the selection is a consuming process where the complexity is related to the amount of the source data. Fan et al. (2017) sought to allocate appropriate training data at different training stages, which helps achieving comparative accuracy with less computational efforts compared with the model trained on the entire data.

Another line of research related to this work is the representation adaptation across domains (Mansour et al., 2008), which is the main strategy of domain adaptation for neural network models. Compared to the aforementioned studies, the proposed approach in this paper combines dynamic data selection and transferable representation learning in a unified RL framework, and is conducted in an effective way on data batches. Especially different from previous work that requires a predefined selection number, our approach not only ensures the selection of related instances in a dynamic manner, but also effectively controls negative transfer (Rosenstein et al., 2005) with limited noise.

## 6 CONCLUSION

In this paper, we proposed a general data selection framework for domain adaptation via reinforcement learning. In detail, the framework matches the representations of the selected data from the source domain and the guidance set from the target domain and pass their similarity in different time steps as rewards to guide the selection distribution generator. Through the generator, different instances from the source domain are selected to train the task-specific predictor. Experiments were conducted on POS tagging, dependency parsing, and sentiment analysis. Results from all tasks demonstrate that our approach outperforms different baselines, especially the one trained on all source data. Ablation study was also provided in analyzing the factors that affects the performance of our approach. Investigations on model convergence, selection numbers, as well as distribution visualizations confirm the validity and effectiveness of our approach.

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

## A  INSIGHT OF REWARD FUNCTION DESIGN

In this section, we will briefly illustrate the motivation to formulate the one-step reward function as Eq. (1). The main idea is to reduce the variance for exploring a path in the state space. By summing up the discounted reward in a exploring path, we can obtain the value function:

$$
\begin{aligned}
V(s) = \sum_{j=1}^{N} \gamma^{j-1} r(s_{j-1}, a_j, s_j) &= \sum_{j=1}^{N} \gamma^{j-1} \{ d(\mathbf{\Phi}_{\hat{B}_{j-1}}^{s_{j-1}}, \mathbf{\Phi}_t^{s_{j-1}}) - \gamma d(\mathbf{\Phi}_{\hat{B}_j}^{s_j}, \mathbf{\Phi}_t^{s_j}) \} \\
&= d(\mathbf{\Phi}^{s_0}, \mathbf{\Phi}_t^{s_0}) - \gamma d(\mathbf{\Phi}_{\hat{B}_1}^{s_1}, \mathbf{\Phi}_t^{s_1}) + \gamma d(\mathbf{\Phi}_{\hat{B}_1}^{s_1}, \mathbf{\Phi}_t^{s_1}) \\
&\quad - \dots + \gamma^{N-1} d(\mathbf{\Phi}_{\hat{B}_{N-1}}^{s_{N-1}}, \mathbf{\Phi}_t^{s_{N-1}}) - \gamma^N d(\mathbf{\Phi}_{\hat{B}_N}^{s_N}, \mathbf{\Phi}_t^{s_N}) \\
&= d(\mathbf{\Phi}^{s_0}, \mathbf{\Phi}_t^{s_0}) - \gamma^N d(\mathbf{\Phi}_{\hat{B}_N}^{s_N}, \mathbf{\Phi}_t^{s_N}),
\end{aligned}
$$

where $s_0$ denotes the the final state in last epoch served as an initial state for the current iteration. It shows that the value function for our reinforcement learning process only rely on the initial state and the final state, with a stronger dependency on the initial state as $\gamma < 1$. Such a result complies with the intuition to minimize the current distribution discrepancy between source and target representations, i.e., $d(\mathbf{\Phi}^{s_0}, \mathbf{\Phi}_t^{s_0})$, as the objective, which implies that different exploring paths with similar initial and final states gives alike total value function for policy updates. Hence we achieve the crucial idea to minimize the trivial discrepancy loss in a reinforced exploring way that allows finding optima closer to the global one and a lower path sampling variance.

