# OpenReview forum: "DOMAIN ADAPTATION VIA DISTRIBUTION AND REPRESENTATION MATCHING: A CASE STUDY ON TRAINING DATA SELECTION VIA REINFORCEMENT LEARNING"
_ICLR.cc/2019/Conference_

### Official Review · AnonReviewer3 · 2018-10-27
**This paper proposes an algorithm that jointly selects training samples for domain adaptation and performs a down stream task such as classification, POS tagging, parsing etc. While the proposed model is interesting and innovative, experimental evaluations are largely lacking and insufficient to establish claims of the authors.**

**Rating:** 5
**Confidence:** 4

**Review:**

The paper aims to address issues with Domain Adaptation by using RL approaches. Domain Adaptation is an actively studied area in NLP research and so this paper is relevant and timely. This paper proposes and algorithm that is in line with work that aims at selecting data smartly when performing Domain Adaptation. The proposed algorithm learns representations for text in the source and target domains jointly. The proposed algorithm has two components i) a selection distribution generator (SDG) and ii) a task specific prediction for tasks being POS tagging, Dependency parsing and Sentiment Analysis.

While the proposed algorithm is interesting from a RL perspective and make sense, there is no explanation provided as to why this algorithm should do better over non RL based approaches for tasks such as Sentiment Analysis.

Domain Adaptation is widely studied for Sentiment Analysis and a lot of current research focuses on the various aspects of domain data, such as word and sentence level semantics, when developing algorithms. For example the following papers all (saving the third) address the problem of Domain Adaptation for Sentiment Analysis through various approaches fairly similar to the authors' algorithm, that provide similar if not better results than those provided in the paper,

[1]Barnes, Jeremy, Roman Klinger, and Sabine Schulte im Walde. "Projecting Embeddings for Domain Adaption: Joint Modeling of Sentiment Analysis in Diverse Domains." arXiv preprint arXiv:1806.04381 (2018).
[2] Ziser, Yftah, and Roi Reichart. "Pivot Based Language Modeling for Improved Neural Domain Adaptation." In Proceedings of the 2018 Conference of the North American Chapter of the Association for Computational Linguistics: Human Language Technologies, Volume 1 (Long Papers), vol. 1, pp. 1241-1251. 2018.
[3]An, Jisun, Haewoon Kwak, and Yong-Yeol Ahn. "SemAxis: A Lightweight Framework to Characterize Domain-Specific Word Semantics Beyond Sentiment." arXiv preprint arXiv:1806.05521 (2018).

Particularly the second paper is a clear improvement over SCL (the earlier pivot based approach), a baseline that is considered by the authors in this work. There are no comparisons against this work in this paper, yet the authors compare against SCL alone.

Due to lack of comparisons against state-of-the-art in Sentiment Analysis/Domain Adaptation for Sentiment Analysis it is hard to accept the claims made by the authors on the superiority of their algorithm. Had their paper aimed at improving over other RL based approaches for Domain Adaptation for Sentiment Analysis, some experiments could be over looked.

But, when making a claim that addresses the problem of Sentiment Analysis, comparisons against the state-of-the-art non RL based approaches is extremely important. Particularly, given the size of the data sets used, one could use lexical/dictionary based approaches [3] and improve upon the classification accuracies without having to train such an involved algorithm.

Furthermore there is no qualitative analysis provided to gain insights into the behavior of the embeddings spaces of the target and source domains that are learned jointly via the proposed algorithm. At least such an analysis would have provided some insight into why the authors' RL based solution is better than a non RL based solution.

The lack of reference or comparisons against relevant literature is future highlighted by the seemingly relevant, yet largely dated related works section.

---

### Official Review · AnonReviewer2 · 2018-11-02
**Interesting approach for learning domain-invariant features + dataset selection**

**Rating:** 7
**Confidence:** 3

**Review:**

== Originality ==
The idea of matching features/representations across the source domain and target domain is an old idea, but it is executed in an interesting new way in this paper.

In this approach, feature representations are learned by training a neural classifier on the source domain, and an RL agent influences the feature representation by iteratively adding/removing examples from the source training data. The RL agent receives reward when the resulting feature representation causes the source domain data and target domain data to look more similar in distribution in feature space. To efficiently estimate the improvement in feature matching, a nice data bucketing strategy is used.

The novelty of the approach is the main strength of this paper.

== Quality of results ==
The experimental results seemed overall positive, but I felt that they could have been stronger.

For POS tagging, the authors don't compare against the domain adaptation methods mentioned in their related work section. Instead, they compare against Bayesian optimization using several heuristic criteria, and it was unclear where this baseline comes from. This made it hard to see whether the new approach represents a true improvement over existing techniques.

For dependency parsing, it appears that the proposed approach is outperformed by simply training on all of the source domain data.  It would be interesting to know whether this is because feature-matching is not a good proxy for target domain performance (objective mismatch) or whether the RL system converged to a poor local optima (optimization failure).

== Clarity ==
I felt that the abstract and introduction were vague in describing the main conceptual contribution.

However, Section 2 (The Approach) was clearly written and I came away understanding exactly what the authors are doing.

== Minor comments ==
- Algorithm 1 seems to have a typo: the definition of \nabla \tilde{J}(W) on the second to last line seems to be missing \nabla \log \pi
- Many citations throughout the paper need to be wrapped in parentheses

== Conclusion ==
This paper presents an interesting new approach for dataset selection and learning domain-invariant representations.

Pros:
- originality of the approach

Cons:
- Experiments could have been more convincing:
    - should compare against at least one other state-of-the-art domain adaptation method
    - results on dependency parsing (the most challenging task they consider) were mostly negative
    - evaluation on other more recent multi-domain NLP tasks would have been nice (e.g. MultiNLI)
- Abstract and intro could provide better description of the conceptual contribution, as well as motivation

---

### Official Review · AnonReviewer1 · 2018-11-03
**Promising experiments, but I was confused about the method details and motivations**

**Rating:** 4
**Confidence:** 2

**Review:**

Response to author comments:

Unfortunately I am still significantly unclear on why RL is useful here.  The author response attempts to clarify that by pointing me to paragraph 2 of the intro, which states that RL has been used for data selection in other settings in the past.  What would help me (and I believe, the paper) more is a reason why greedy selection isn't sufficient for this particular problem.  Even just a single motivating example would be extremely helpful.  R3 mentioned similar concerns in their review, saying that the paper lacks explanation for why RL would win over non-RL for e.g. sentiment analysis.

Likewise, while I appreciate the authors comparing against a stronger baseline in Figure 3, I don't know how to interpret the figure.  Why is Figure 3(b) better than Figure 3(c), and why does using RL cause that difference to arise?

Original review:

Domain adaptation is an interesting task, and new methods for it would be welcome.  This paper appears to have technical depth and the experimental results are promising.  However, the presented approach is complex, and I found it very hard to understand -- both in terms of how exactly it works, and in terms of why the chosen techniques were chosen.  More detail on my questions and confusions follows.

First, I never understood the motivation for using RL here.  If minimizing the distance between selected data from the source domain and data in the target domain is the objective (equation 1), how does RL help?  The reward seems like it is immediate in each time step.  How does the *order* in which we add source examples to our collection matter?  I never understood the crucial difference that made the RL approach outperform the baselines that just select examples that minimize e.g. JS divergence.  Neither the paper's discussion of motivation nor the experimental analysis clarifies this.

The paper says in Section 2.1 that a formal description of the representations is to follow.  I didn't see this description (I do not see a formal definition of how the feature extractor works, and e.g. how it produces vectors that are *distributions* that can be used within e.g. JS divergence).

The paper also says it follows (Ruder and Plank, 2017) in using JS as a baseline, but as I understand that work the JS baseline is computed over words, not learned representations.  What is done in the submission, is the JS baseline over words in the instances, or the representations from the feature extractor?

What is the reason for partitioning the source data into disjoint "data bags"?  Why not just select the best source domain examples (from among all the source data) using RL?

The experiments are generally over enough tasks and compare against several baselines, and although the empirical wins are not that large I feel that they would be sufficient for publication if not for my other concerns.  The analysis (sec 4) did not make it clear to me why the RL approach works.  The visualization in Figure 3 only contrasts the proposed approach with a weak baseline of selecting all source data -- what we really need to see is an analysis that reveals why the learning of a policy with RL is better than simply greedily minimizing JS for each source data selection, for up to some limit of n selections.

Minor
The paper has a number of typos
The citations in the paper are mis-formatted -- seem to use shortcite where they shouldn't (e.g. "scenarios Akopyan and Khashba (2017)" should be "scenarios (Akopyan and Khashba, 2017)").
When the policy \pi_w(a | s) is introduced at the start of Sec 2.2, it uses symbols (a, s) that have not been defined, also that policy variable is not really utilized in the text so it could be deleted.

---

### Public Comment · (anonymous) · 2018-12-10
**Similar work exists**

Similar work on using Reinforcement learning for sample selection published (http://www.ecmlpkdd2018.org/wp-content/uploads/2018/09/81.pdf)  and need to be referred.

---

> ### Author Response · Authors · 2019-01-25
> **Thanks for your kind remainder**
>
> Thank you for your information. We will carefully read the paper and refer to them in the future version.

---

### Meta-Review · Area_Chair1 · 2018-12-14
**A potentially interesting idea but 2/3 reviewers share strong concerns about the empirical results and overall clarity of the paper.**

**Confidence:** 4
**Recommendation:** Reject

**Metareview:**

This paper investigates a data selection framework for domain adaptation based on reinforcement learning.

Pros:
The paper presents an approach that can dynamically adjust the data selection strategy via reinforcement learning. More specifically, the RL agent gets reward by selecting a new sample that makes the source training data distribution closer to the target distribution, where the distribution comparison is based on the feature representations that will be used by the prediction classifier. While the use of RL for data selection is not entirely new, the specific method proposed by the paper is reasonably novel and interesting.

Cons:
The use of RL is not clearly motivated and justified (R1,R3) and the method presented in this paper is rather hard to follow might be overly complex (R1). One fair point R1 raised is more clean-cut empirical evaluation that demonstrates how RL performs clearly better than greedy optimization. The authors came back with additional analysis in Section 4.2 to address this question, but R1 feels the new analysis (e.g., Fig 3) is not clear how to interpret. A more thorough ablation study of the proposed model might have addressed the reviewer's question more clearly. In addition, all reviewers felt that baselines are not convincingly strong enough, though each reviewer pointed out somewhat different aspects of baselines. R3 is most concerned about baselines being not state-of-the-art, and the rebuttal did not address R3's concern well enough.

Verdict:
Reject. A potentially interesting idea but 2/3 reviewers share strong concerns about the empirical results and overall clarity of the paper.